# Structure and Properties of a Metallocene Polypropylene Resin with Low Melting Temperature for Melt Spinning Fiber Application

**DOI:** 10.3390/polym11040729

**Published:** 2019-04-22

**Authors:** Renwei Xu, Peng Zhang, Hai Wang, Xu Chen, Jie Xiong, Jinpeng Su, Peng Chen, Zhicheng Zhang

**Affiliations:** 1Lanzhou Petrochemical Research Center, Petrochina, Lanzhou 730060, China; xurenwei@petrochina.com.cn (R.X.); zhangpeng931@petrochina.com.cn (P.Z.); chenxu001@petrochina.com.cn (X.C.); 2Xi’an Key Laboratory of Sustainable Energy Materials Chemistry; Department of Applied Chemistry; School of Science; Xi’an Jiaotong University; Xi’an 710049, China; xiongjiely@stu.xjtu.edu.cn; 3Ningbo Key Laboratory of Polymer Materials, Ningbo Institute of Materials Technology and Engineering (NIMTE), CAS, Ningbo 315201, China; sujinpeng@nimte.ac.cn; 4Center of Materials Science and Optoelectronics Engineering, University of Chinese Academy of Sciences, Beijing 100049, China

**Keywords:** metallocene isotactic polypropylene, melt-spinning, polypropylene fibers

## Abstract

An isotactic polypropylene (iPP-1) resin with low melting temperature (*T*_m_) is synthesized by a metallocene catalyst and investigated for melt-spun fiber applications. The structure, thermal and mechanical properties, and feasibility of producing fibers of a commercial metallocene iPP (iPP-2) and a conventional Ziegler–Natta iPP (iPP-3) are carefully examined for comparison. *T*_m_ of iPP-1 is about 10 °C lower than the other two samples, which is well addressed both in the resin and the fiber products. Besides, the newly developed iPP-1 possesses higher isotacticity and crystallinity than the commercial ones, which assures the mechanical properties of the fiber products. Thanks to the addition of calcium stearate, its crystal grain size is smaller than those of the two other commercial iPPs. iPP-1 shows a similar rheological behavior as the commercial ones and good spinnability within a wide range of take-up speeds (1200–2750 m/min). The tensile property of fibers from iPP-1 is better than commercial ones, which can fulfill the application requirement. The formation of the mesomorphic phase in iPP-1 during melt spinning is confirmed by the orientation and crystallization investigation with wide angle X-ray diffraction (WAXD), which is responsible for its excellent processing capability and the mechanical properties of the resultant fibers. The work may provide not only a promising candidate for the high-performance PP fiber but also a deep understanding of the formation mechanism of the mesomorphic phase during fiber spinning.

## 1. Introduction

The majority of nonwoven polyolefin fibers are of great commercial interest because of their unique properties, including hydrophobicity, good mechanical properties, and excellent chemical resistance [1,2]. Among them, fibers fabricated from isotactic polypropylene (iPP) are one of the lightest and most widely utilized. Traditional commercial iPP is generally polymerized with a heterogeneous titanium-based Ziegler–Natta catalyst, whose tacticity, molecular weight (M_w_), and polymer dispersity index (PDI) are tightly dependent on the catalysts and the polymerization conditions [3,4]. The development of metallocene catalytic systems offers new access for both the isospecific polymerization of olefins and the new polymer microstructures thanks to the single-site nature of metallocene catalysts. The metallocene polyolefins exhibit interesting physical properties, which are not attainable with conventional Ziegler–Natta catalysts [5]. In comparison with the Ziegler–Natta-catalyzed polypropylene (PP), metallocene polypropylenes are uniform polymers with precisely controlled M_w_, end groups, stereoregularity, and narrow PDI [6]. Those characters make metallocene-based iPP a great candidate for producing high-quality industrial fibers by melt spinning.

Crystallization plays a crucial role in determining the structure of not only the iPP resin but also the resulting fibers [7]. Investigations on crystalline structures have revealed that the crystalline phase of iPP fibers is usually composed of the *α* phase crystal. The *α* form, which can be easily obtained by the crystallization of polymer in melts or solutions, is the most stable and the most common form in different iPP products [4]. The *β* form, which is produced during the crystallization of a sheared melt, crystallization in a temperature gradient, or during the addition of certain nucleating agents, has to be achieved under well-controlled conditions in iPP products [8]. The *γ* phase is usually formed at high pressure and favored by the low-molecular-weight fraction of polydisperse melts [9,10]. In addition to these crystal phases, a mesomorphic phase could be obtained at high cooling rates, which is usually formed during high-speed spinning [11,12]. The mesomorphic phase is characterized by an order intermediate between those found in crystalline and in amorphous phases. However, the structure of this phase has been a matter of debate since its initial report. The mesomorphic phase is composed of bundles of parallel chains, which are typically observed in all polymorphic forms of the threefold helical conformation of PP. Bundles are terminated in the direction of the chain axis by helix reversals or other conformational defects [13]. In the bundles, long-range ordering is maintained only along the chain axes, whereas a large amount of disorder is present in lateral packing [3].

The melt spinning of iPP into fibers has been intensively studied during past decades [14,15,16]. The fiber structure is mainly formed during crystallization, which starts below the spinneret orifices, in the cooling zone through the solidification of an extruded stream. During fiber formation, crystallization occurs under nonisothermal conditions and high tensile stress, and the cooling rate and tensile stress strongly affect the crystallization of iPP [17]. At low take-up speeds, the cooling rate and the applied tensile stress are low, and the crystallization proceeds at relatively high temperatures, which favors the formation of a monoclinic α crystal in as-spun fibers. As the take-up speed increases, the cooling rate is enhanced, and the crystallization temperature decreases. Structures with high mesomorphic phase content are formed in major during crystallization under such conditions [18,19,20].

The melt-spinning of metallocene catalyst iPP has been investigated in the last period, Bond and Spruiell demonstrated that the molecular weight and molecular weight distribution (MWD) of metallocene iPP resins had a significant influence on the development of the structure and properties of fibers prepared from these resins. Compared to the Ziegler–Natta-catalyzed resins with similar average molecular weight, the narrower MWD of metallocene resins appeared to be the primary factor that produced differences in the structure and properties of fibers spun from these resins [21]. In addition, the effect of the spinning speed on the density of fibers, crystallinity, and crystalline and non-crystalline orientation function was investigated. It was found that the density, birefringence, tensile strength, and crystalline and non-crystalline orientation function could be generally improved with increases in the spinning speed. The metallocene iPP fibers were identified to have a breaking tensile strength up to 50% higher than of the Ziegler–Natta iPP fiber at similar spinning speeds [22]. Therefore, developing metallocene iPP for fiber applications may significantly improve the fiber strength and extend their applications. 

It has been reported that the crystallinity and melting temperature of metallocene-catalyzed iPP could be lower than those of conventional iPP fibers, which could be ascribed to the region-defects in the 2,1 and 3,1 insertions. At lower melting temperature, there exist more “tie” molecules between the PP crystallites, which lead to the enhanced tensile strength and elongation at break of metallocene-catalyzed iPP fibers [6,23]. Therefore, the metallocene iPP with lower melting temperature is a promising polymer for fiber production. In this work, we present a low-melting-temperature metallocene iPP for melt spinning fiber application. The results from a commercial Ziegler–Natta iPP and a metallocene iPP are given for comparison. The structure, thermal properties, and mechanical properties of the three types of iPP are primarily tested. The iPP fibers are fabricated from three iPPs via melt spinning following the same process, and the tensile properties, orientation, and crystallization of the prepared iPP fibers are carefully investigated. The *T*_m_ metallocene iPP exhibits remarkable advantages in both the processing and the mechanical performance of the resultant products.

## 2. Materials and Methods 

### 2.1. Materials

Isotactic polypropylene sample 1 (iPP-1) is synthesized from C_2_-symmetric metallocene catalysts (rac-Me_2_Si(2-Me-4-Ph-Ind)_2_ZrCl_2_) [24] with a M_w_ of 147,000 and PDI of 2.5, which is a pilot product synthesized by the China Petroleum Lanzhou Chemical Research Center (Lanzhou, China). Calcium stearate at 500 ppm (Guangdong Wei Lin Na New Material Technology Co. Ltd., Guangzhou, China, ≥98%), 750 ppm primary antioxidant (Irganox 1010), and 1500 ppm secondary antioxidant (Irgafos 168) (Basf Performance Chemicals (Shanghai) Co. Ltd.) were added into the iPP powder to prepare pellets via extrusion. Isotactic polypropylene sample 2 (iPP-2) is a commercial metallocene PP purchased from Lyondell Basell Industries (Basell HM562S, Rotterdam, Netherlands) with a M_w_ of 157,000 and PDI of 2.9. Isotactic polypropylene sample 3 (iPP-3) is a commercial Ziegler–Natta PP produced by Exxon Mobil Corporation (Exxon 3155E3) with a M_w_ of 157,000 and PDI of 2.4. All the samples are used as received without any further purification unless notified. 

### 2.2. Instrumentation and Characterization

^13^C NMR spectra were recorded on a Bruker (Advance III) 400 MHz spectrometer with 1,2-dichlorobenzene-*d*_4_ as the solvent, at 120 °C. Gel permeation chromatography (GPC) was performed on a Waters GPC 1515 HPLC pump (Waters, Milford, MA, USA) and Wyatt DAWN HELEOS-II light scattering/Wyatt Optilab Rex refractive index detectors with 1,2,4-Trichlorobenzene as the eluent and polystyrene as an internal standard, at 135 °C. Differential scanning calorimetric (DSC) analysis was performed on a Netzsch DSC 200 PC (Netzsch, Free State of Bavaria, Germany) in a nitrogen atmosphere at a scanning rate of 15 °C min^−1^ from 30 to 180 °C and the results were collected from the second heating circle. The melt flow rate (MFR) analysis was performed on a Melt Flow Testers 7028 (Ceast, Turin, Italy) according to the GB/T 3682-2000. Tensile properties, bending properties, and impact strength analysis were performed on a universal testing machine 5566 according to the China-National Standard System (GB/T 1040-2006, GB/T 9341-2008 and GB/T 1843-2008, respectively). The crystallization property was performed on a polarizing microscope (POM) DM2500P (Leica, Solms, Germany). The polarized optical microscopic photographs were obtained by using a German Leica DMLP microscope. The temperature-rising elution fractionation (TREF) profiles were generated using Polymer Char 300 TREF (Polymer Char, Valencia, Spain) in a nitrogen atmosphere with 1,2,4-Trichlorobenzene as the solvent. The rheological behavior was studied at 230 °C by using a capillary rheometer (GOTTFERT, RHEO-TESTER 2000, Berlin, Germany) equipped with a capillary die (D = 0.5 mm, L/D = 60). The capillary die was sufficiently long so that the Bagley correction was not applied while the Rabinowitsch correction was applied. Crystallization and orientation were evaluated using a 2D wide angle X-ray diffraction (WAXD) (D8 DISCOVER, Bruker, Karlsruhe, Germany) under a voltage of 40 kV and a current of 40 mA with Cu Kα radiation (λ = 0.154 nm). Thermal analysis was carried out on a differential scanning calorimeter (DSC-I, Mettler Toledo, Switzerland) with samples of ca. 5 mg sealed in aluminum pans heated from 50 to 220 °C at 10 °C min^−1^ in a nitrogen atmosphere.

### 2.3. Melt-Spinning and Hot-Drawing

The melt-extrusion and spinning were conducted by using a single-screw extruder equipped with a 2.4 mL/rev metering-pump and an eight-hole spinneret, and the diameter of each hole was 0.6 mm. The extrusion temperature was set at 215, 265, 265, and 265 °C from the feed zone to the end of the extruder, respectively, and the spinneret temperature was set at 265 °C. The as-spun fibers were taken up at 1200, 2100, and 2750 m/min, respectively. The as-spun fibers were then hot-drawn by passing through five rollers (D = 125 mm) and two heat plates (L = 500 mm) between the 3rd and 4th rollers and the 4th and 5th rollers, respectively. The 2nd, 3rd, and 4th rollers were heated to 70, 130, and 145 °C, respectively, while the 1st and 5th rollers were at room temperature. Both the heat plates were kept at 130 °C. For comparison, the winding speed was fixed at 200 m/min during drawing, and the total draw ratio (DR) was fixed at 1.6, which was the maximum achievable DR for the fibers spun at the highest speed, i.e., 2750 m/min.

## 3. Results and Discussion

### 3.1. Microstructural Characterization

The microstructures of the iPP samples (Table 1) are determined with solution ^13^C nuclear magnetic resonance (NMR) spectroscopy analysis. All of the samples are fully regioregular, and no or negligible amounts of regiomistakes are detected in their ^13^C NMR spectra. They contain stereoregularity defects, which correspond to isolated *rr* triads, and the number of *rr* defects depend on the catalyst and polymerization temperature. In Table 1, the contents of the isotactic pentad fraction, namely, *mmmm*, in the three iPP samples from iPP-1 to iPP-3 are 98.98%, 97.10%, and 93.64% respectively. The highest meso diad content of 98.98% is observed in iPP-1, which means iPP-1 possesses the highest isotacticity. More or less other types of pentad fractions could be observed in all the three iPP samples. For iPP-1, only two kinds of defect configurations, namely, *rmmr* and *mmrr*, could be observed. In comparison with iPP-1, the other two samples have more defect configurations with higher contents. In particular, a high concentration of the fully syndiotactic *rrrr* pentad is observed in iPP-3. Traditional commercial iPP generally is prepared with the heterogeneous Ziegler–Natta catalytic systems that are composed of multiple sites, and each site may produce polymers with different tacticity. That may address the lower tacticity obtained in the resultant polymer. Different from the Zigler-Natta catalysts, metallocene catalysts are well recognized as single-site systems, which are responsible for the produced polymers with small PDI and high tacticity. For the degradable grade iPP-3 produced by Exxon, the tacticity is much lower than the other two products and more defects could be detected including the highest *mmmr* and *mmrr* sequences among the three samples. Even *rmrm* and *rrrr* sequences could be found in iPP-3, which is absent in the other two samples. The new metallocene iPP produced in the present work possesses the significant isotacticity advantage over the other two commercial products, which is rather important for the crystallization and properties of the resultant fibers.

### 3.2. Structural Characterization

iPP can generally be crystallized in three different polymorphic forms (*α*, *β*, and *γ* forms) characterized by chains in a 3-fold helical conformation, which can be clearly identified by the distinctive reflection in the WAXD scan. The WAXD patterns of several samples for each iPP are shown in Figure 1. We recall that *α* and *γ* forms of iPP present similar X-ray diffraction (XRD) profiles and mainly differ in the position of the third strong diffraction peak, which appears at 2θ = 18.6° ((130)*_α_* reflection) in the α form and at 2θ = 20.1° ((117)*_γ_* reflection) in the *γ* form. Apparently, the crystalline fractions are mainly crystallized in the *α* form in all of the iPP samples as indicated by the presence of the (130)*_α_* reflection at 2θ = 18.6° of the *α* form and the absence of the (117)*_γ_* reflection at 2θ = 20.1° of the *γ* form in the XRD profiles.

The thermal analysis of the PP samples was conducted via differential scanning calorimetry (DSC), and the DSC heating curves are shown in Figure 2. All of the samples exhibit broad melting endotherms characterized by a single peak. The iPP-1 has a *T*_m_ of 10 °C lower than the other two samples (Table 1), independent of the processing conditions. That confirms the reported results that metallocene iPP possesses lower *T*_m_ than the other iPPs such as those catalyzed with the Ziegler–Natta catalyst. The crystallinity of the three types of iPP resins is estimated on the basis of the DSC results by using the following equation, *χ*_c_ = Δ*H_f_*/Δ*H_f_^*^*, where Δ*H_f_* and Δ*H_f_^*^* (145 J/g) [25] refer to the melting enthalpies of the iPP resins and iPP with 100% *α* form crystallinity, respectively. As listed in Table 2, the crystallinity of iPP-1 (about 51.2%) is higher than the other iPP resins, which agrees well with the isotacticity results as discussed above. Further, the metallocene catalyzed iPP may possess different types of microstructural defects, such as stereo-defects and region-defects in the polymer chain. Moreover, the traditional Ziegler–Natta catalyst polypropylene have stereo-defects only. The stereo-defects are mainly *rr* isolated triads, while the region-defects are mainly 2,1 and 3,1 insertions [26,27]. The insertion of regio-irregular units in the main chain leads to a random distribution of the microstructural defects. These defects lower the thickness of the lamellar crystal of the polymer chain, which leads to the metallocene iPP (iPP-1) having lower melting temperatures [28].

Figure 3 shows the polarized optical microscopy (POM) photographs of the three iPP samples. The commercial iPP (Figure 3b,c) reveals a common spherulitic structure of the *α* crystal form. However, iPP-1 shows a distorted spherulite shape with a much smaller crystal domain size. That might be ascribed to the addition of calcium stearate to remove the residue acid, which can serve as the nucleating agent for the crystallization of PP as well. As shown in the Appendix A, as the temperature decreases, the crystallization is firstly observed in iPP-2 and iPP-3 at 150 °C, while iPP-1 starts to crystallize at 140 °C. Meanwhile, once the crystal nucleus is formed in iPP-2 and iPP-3, the crystal grows gradually along with the slow formation of the new nucleus. However, in iPP-1, a great number of nuclei are formed quickly right from the beginning, and all the nuclei grow simultaneously. The quick formation and growth of the crystal domains are due to the addition of calcium stearate as a nucleating agent resulting in the distorted spherulite domains.

Temperature rising elution fractionation (TREF) is currently the best technique to obtain the composition distribution of polymers with high crystallinity. The difference in the samples is observed in their TREF fractograms as indicated in Figure 4. First of all, the percentages of the soluble fraction eluted at room temperature are 0.3% for iPP-1, 0.9% for iPP-2, and 1.3% for iPP-3 (Table 3). That can be ascribed to the largest iPP-1 isotacticity. For both iPP-2 and iPP-3 samples, the major part of the polymer is eluted at 121 °C, while iPP-1 is mainly eluted at 115 °C. That could be due to the lower *T*_m_ of iPP-1 compared to the other two samples.

### 3.3. Mechanical Properties

The mechanical properties of the iPP samples are tested and summarized in Table 4. The results indicated that the tensile strength and flexural modulus of the three iPP samples are rather close. The iPP-1 sample exhibits the highest simply supported beam impact strength of 2.15 kJ/cm^2^ and the highest elongation at break of 16.21% because the iPP-1 sample has the highest isotacticity and crystallinity.

### 3.4. Processing Properties

Factors affecting the processing properties of PP mainly include the melt flow rate, ash content, and rheological property, and the results of the three samples are shown in Figure 5 and Table 5. In Figure 5, at low-frequency (100–500 Hz), the viscosity of the iPP samples is decreasing in the order of iPP-2 > iPP-1 > iPP-3. The lower viscosity of iPP-1 compared to iPP-2 could be attributed to the lower *T*_m_ of iPP-1. The higher viscosity of iPP-1 compared to iPP-3 might be ascribed to the larger isotacticity and crystallinity of iPP-1 compared iPP-3. As the frequency increases, the viscosity of all the samples shows an invisible difference. The production of PP nonwoven fabric requires a melt flow rate between 30 and 40 g 10/min. As shown in Table 5, all iPP samples with MFR from 34–36 g 10/min meet that requirement. The ash content remarkably influences the stability of the spinning, and the results indicated that all of the PP samples could satisfy this requirement, and the iPP-1 sample has the lowest ash content (Table 5).

### 3.5. Fiber Properties

All of the three types of iPP resins were subjected to conventional melt extrusion and spinning, and showed good spinnability within a wide range of take-up speeds (1200–2750 m/min). Tensile tests were performed to evaluate the mechanical properties of the iPP fibers prepared in the present work. The tensile properties of the as-spun and hot-drawn iPP fibers are listed in Table 6 and Table 7, respectively.

In Table 6, the tenacity of the as-spun fiber of iPP-1 constantly decreases from 1.69 cN/dtex to 1.22 cN/dtex as the take-up speed increases from 1200 to 2750 m/min. However, for the other two samples (iPP-2 and iPP-3), the tenacity decreases initially as the take-up speed increases from 1200 to 2100 m/min and increases subsequently as the take-up speed further increases to 2750 m/min. Such inconsistencies are also observed in the trend of modulus varying with the take-up speed for the three types of samples (Table 6). At a given take-up speed, the as-spun fibers prepared using iPP-1 show lower tenacity and modulus than those of the other two samples in most cases. The elongation at break of all the samples decreases constantly as the take-up speed increases. At a given take-up speed, the as-spun fibers prepared using iPP-1 show intermediate elongation at break between the two other types of the samples.

For the fiber samples, tenacity is dominated mainly by molecular orientation, whereas modulus is dominated mostly by structural perfection (orientation and crystallization). During melt spinning, multiple factors control molecular orientation and structural perfection, manifested by various parameters, such as tension, strain rate, and cooling rate in the thread line. Although all of these parameters increase as the take-up speed increases, different impacts are observed. For instance, the increased tension and the strain rate may promote molecular orientation, but the increment in the cooling rate may freeze the molecular chains too rapidly to provide enough time and mobility for their orientation and crystallization. We attempt to explain the observed variations in fiber properties in the succeeding discussion by considering the fiber structures developed under various conditions.

In synthetic fiber and textile industries, hot-drawing treatment is usually performed to further improve the properties of fibers and stabilize their fiber. As a general rule, tenacity and modulus increase, but the elongation at break decreases as the draw ratio (DR) increases. A low take-up speed for the as-spun fibers indicates a highly achievable DR during hot drawing. In the present work, we chose a low and constant DR of 1.6 to ensure a practical drawing process for the as-spun fibers and to fairly compare them.

In Table 7, the tenacity of the drawn fiber is significantly increased compared to the as-spun fiber. The modulus of the drawn fibers is also increased in most cases. The line density and the elongation at break of the drawn fibers are notably decreased. These changes are consistent with the general rule of fiber drawing. In other words, the fiber becomes thinner, stronger, and more rigid upon hot drawing, and these property values in the present work are acceptable for general textile applications. For the as-spun fibers prepared at low take-up speeds, DR exceeding 1.6 can be readily applied during hot drawing, and fibers with high tenacity and modulus can be produced if needed.

### 3.6. Thermal Properties of Fibers Produced

Thermal properties are essential for fibers to determine the appropriate processing and usage temperature windows. Figure 6 shows the DSC thermograms of the as-spun and drawn fibers during the first heating run. In all the thermograms, distinct endothermic melting peaks that are wide or split and have shoulders are observed, implying that all of the fiber samples are crystalline but have complex crystal forms or morphologies. The iPP is well known to be able to crystallize into multiple forms, such as *α*, *β*, and *γ*, under different conditions. If the cooling rate is high, typically encountered in high-speed spinning, iPP may form a mesomorphic phase with its structure close to the *α* crystal except with an imperfect chain organization. Therefore, having a complex crystalline or mesomorphic structures for the iPP fibers is not surprising because complex thermal and tension histories are experienced during melt spinning and hot drawing.

Melting peaks are separated using *PeakFit* to derive a series of Gaussian subpeaks (Table 8 and Table 9) and to perform a relevant quantitative analysis. Interestingly, the melting peaks of the fibers prepared using iPP-1 can be separated into three subpeaks, whereas those of the two other types of fibers can be divided into two subpeaks. This result may be due to the presence of a nucleating agent (calcium stearate) that induces the formation of small crystals with the lowest *T*_m,3_ (143.8–151.5 °C) in iPP-1. The fibers prepared with iPP-1 have *T*_m,1_ of 159.8–162.4 °C and *T*_m,2_ of 151.3–156.8 °C, which are significantly lower than those of the two other types of samples (iPP-2 and iPP-3) for the lower *T*_m_ of the pristine iPP-1 resin. 

### 3.7. Orientation and Crystallization

2D-WAXD is used as a nondestructive tool to characterize the orientation and crystallization of various fibers. Figure 7 and Figure 8 (and Appendix A) show the 2D-WAXD patterns of the as-spun and drawn fibers, respectively. In Figure 7, diffraction arcs are observed in the patterns of the as-spun fibers prepared at low take-up speeds, thereby transforming into bright and sharp spots at high take-up speeds. This finding suggests that the orientation and crystallization of the as-spun fibers are significantly improved as the take-up speed increases. For the drawn fibers (Figure 8), all of the 2D-WAXD patterns are featured by bright and sharp spots, indicating that these samples are highly oriented and crystallized during hot drawing.

The DSC data show the complex crystalline or the formation of mesomorphic structures of the as-spun and drawn fibers. The 2D-WAXD patterns are integrated into 1D curves (Figure 9, Figure 10, Appendix A) to clarify this condition. Notably, most of the samples exhibit distinct crystalline diffraction peaks at 14.2°, 16.8°, 18.5°, 21.2°, and 21.8° corresponding to the (110), (040), (130), (111), and (041) planes of the *α* crystal, respectively. Two exceptions are observed, that is, the as-spun fibers (iPP-1 and iPP-3) prepared at a low take-up speed (1200 m/min) have two wide peaks at ca. 15° and 21.3° corresponding to the mesomorphic phase. The mesomorphic peaks are close to those of the *α* crystal. As such, the coexistence of the mesomorphic phase and the *α* crystal is possible, especially for the as-spun fibers. By contrast, a less mesomorphic phase should exist in the drawn fibers because it would be transformed into the *α* crystal during hot drawing.

A close examination of the 1D-WAXD curves shown in Figure 9 and Figure 10 (Appendix A) provides further details. First, for the as-spun fibers, the peaks corresponding to the planes of the *α* crystal, especially those of the (040) and (130) planes, intensify and become sharp as the take-up speed increases, suggesting that the formation of the *α* crystal is favored by the high tension at a high take-up speed. Second, for the drawn fibers, the peaks corresponding to the planes of the *α* crystal further become sharp, suggesting a constant perfection of crystallization during hot drawing. Lastly, for the as-spun fibers prepared at a low take-up speed (1200 m/min), the crystallization behavior reflected by the WAXD data is quite different among the three types of fibers. Specifically, iPP-1 and iPP-3 show a relatively slow crystallization behavior and form the mesomorphic phase, whereas iPP-2 exhibits a rapid crystallization behavior and readily produces the *α* crystal.

## 4. Conclusions

In summary, the structure, properties, and feasibility of producing fibers from a newly developed metallocene iPP resin have been carefully tested by comparing them with the commercial products. The influence of crystalline properties on the processing ability along with the mechanical properties of melt spinning fibers are well discussed. The low *T*_m_ and high isotacticity of iPP-1 are responsible for its advantages both in the fabrication conditions and the properties of the fibers. The newly developed iPP is confirmed to be a promising candidate for producing high-performance fibers from the melt spinning process.

## Figures and Tables

**Figure 1 polymers-11-00729-f001:**
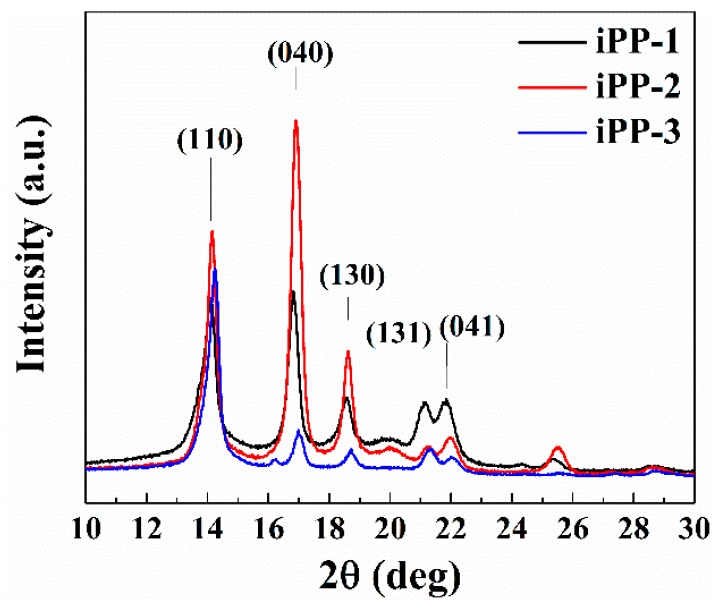
X-ray powder diffraction profiles of as-prepared specimens of the iPP samples.

**Figure 2 polymers-11-00729-f002:**
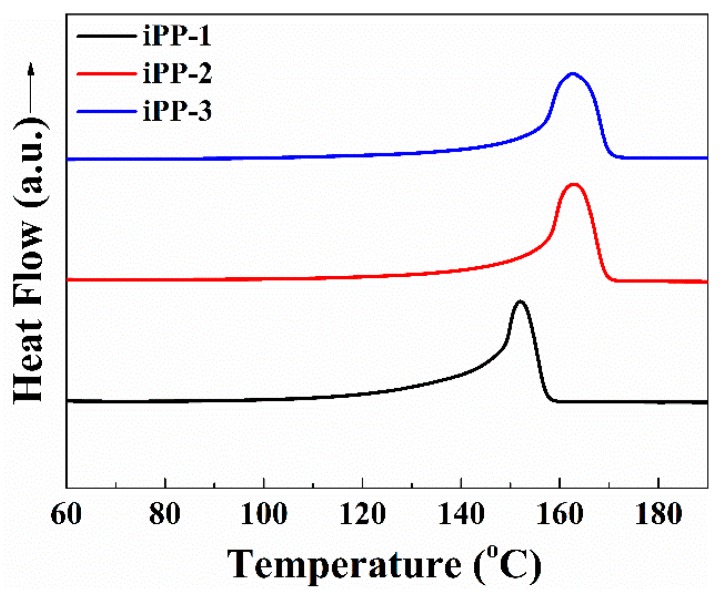
DSC curves of the iPP samples.

**Figure 3 polymers-11-00729-f003:**
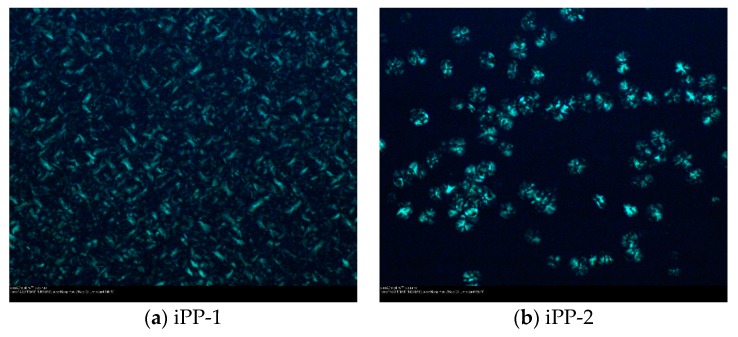
Morphologies of (**a**) iPP-1, (**b**) iPP-2 and (**c**) iPP-3 under a polarizing microscope.

**Figure 4 polymers-11-00729-f004:**
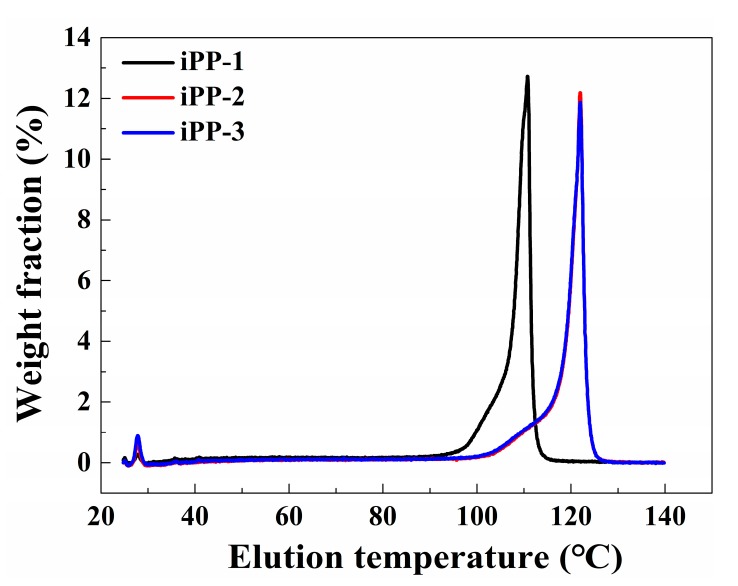
Temperature rising elution fractionation (TREF)curves of the iPP samples.

**Figure 5 polymers-11-00729-f005:**
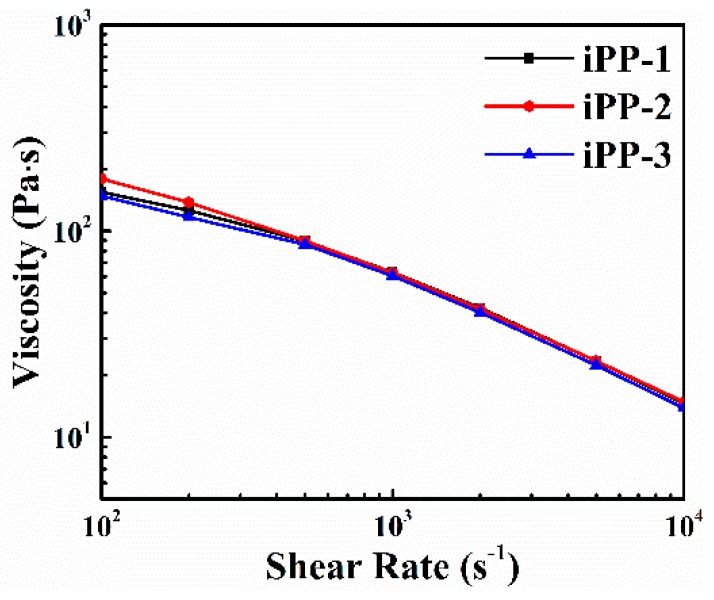
Rheological properties of the iPP samples.

**Figure 6 polymers-11-00729-f006:**
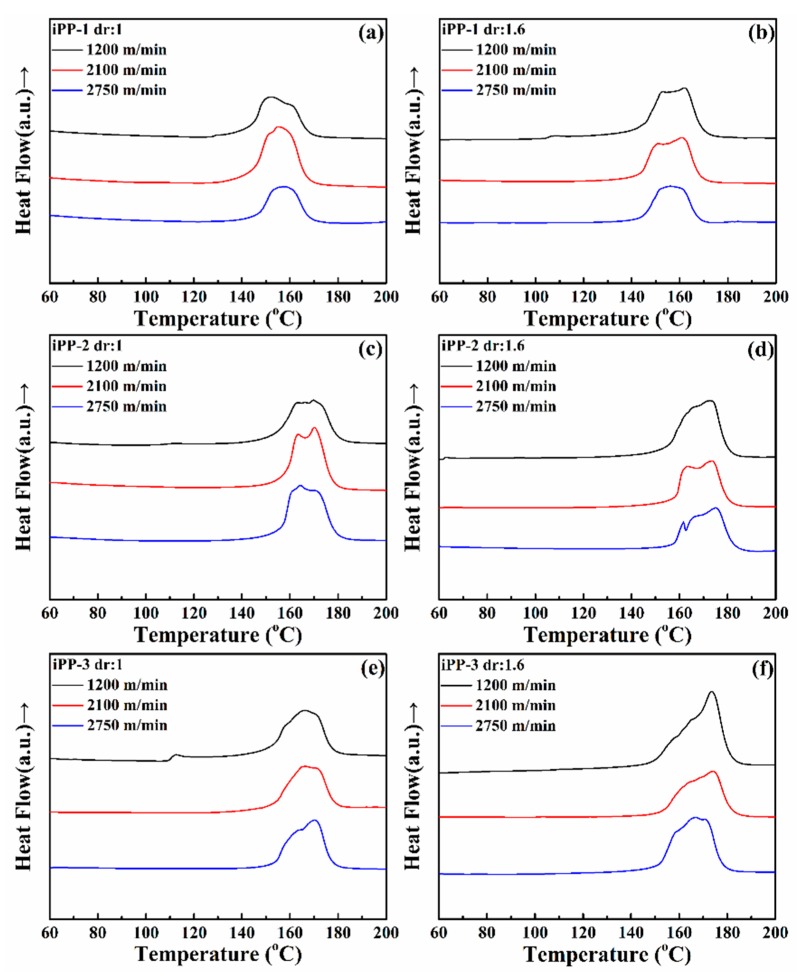
DSC thermograms of (**a**), (**c**), (**e**) as-spun and (**b**), (**d**), (**f**) drawn iPP fibers.

**Figure 7 polymers-11-00729-f007:**
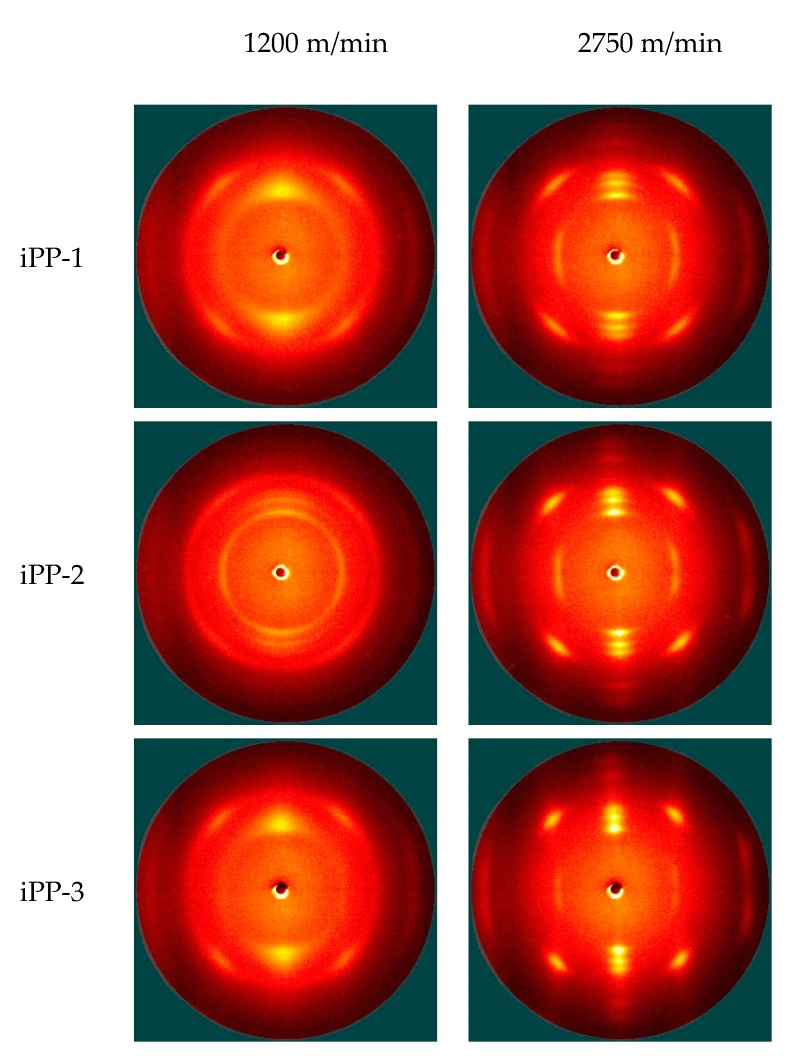
2D-WAXD patterns of as-spun iPP fibers.

**Figure 8 polymers-11-00729-f008:**
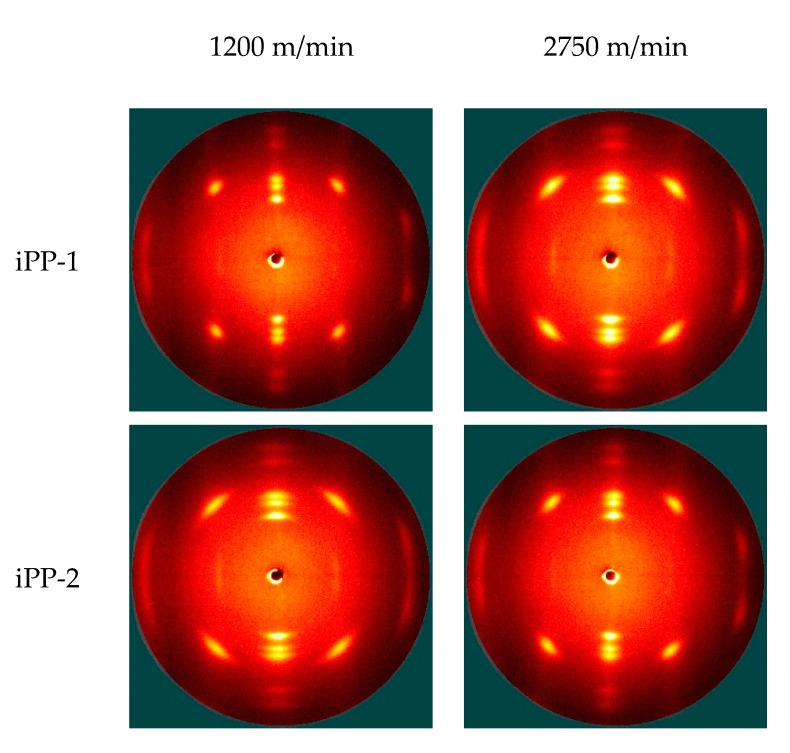
2D WAXD patterns of drawn iPP fibers (draw ratio: 1.6).

**Figure 9 polymers-11-00729-f009:**
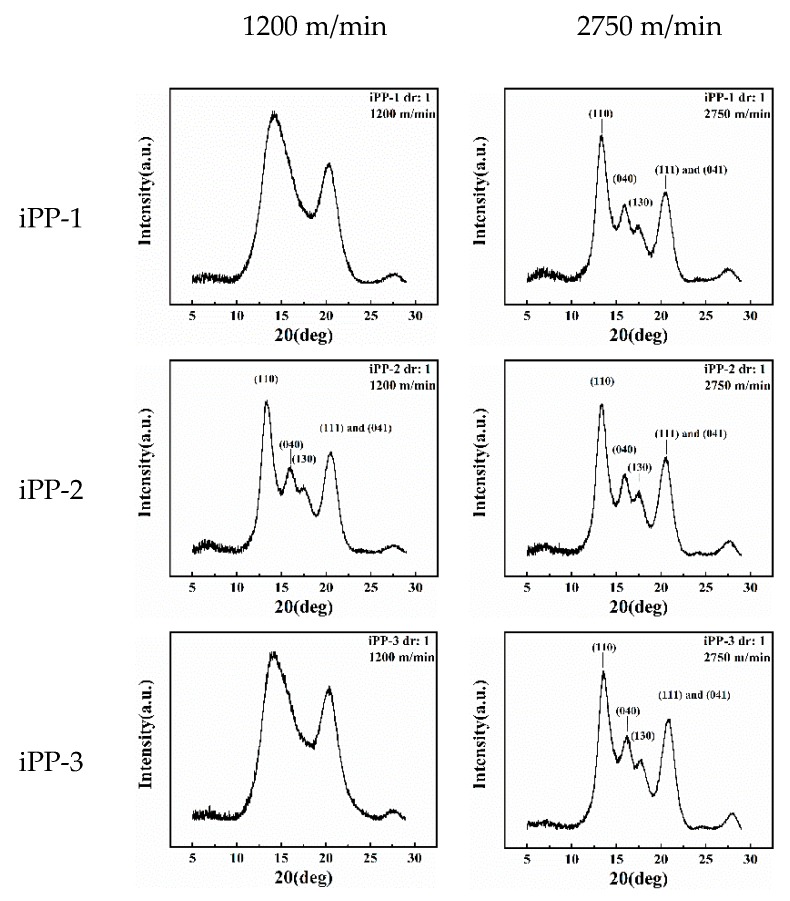
1D WAXD curves of as-spun iPP fibers.

**Figure 10 polymers-11-00729-f010:**
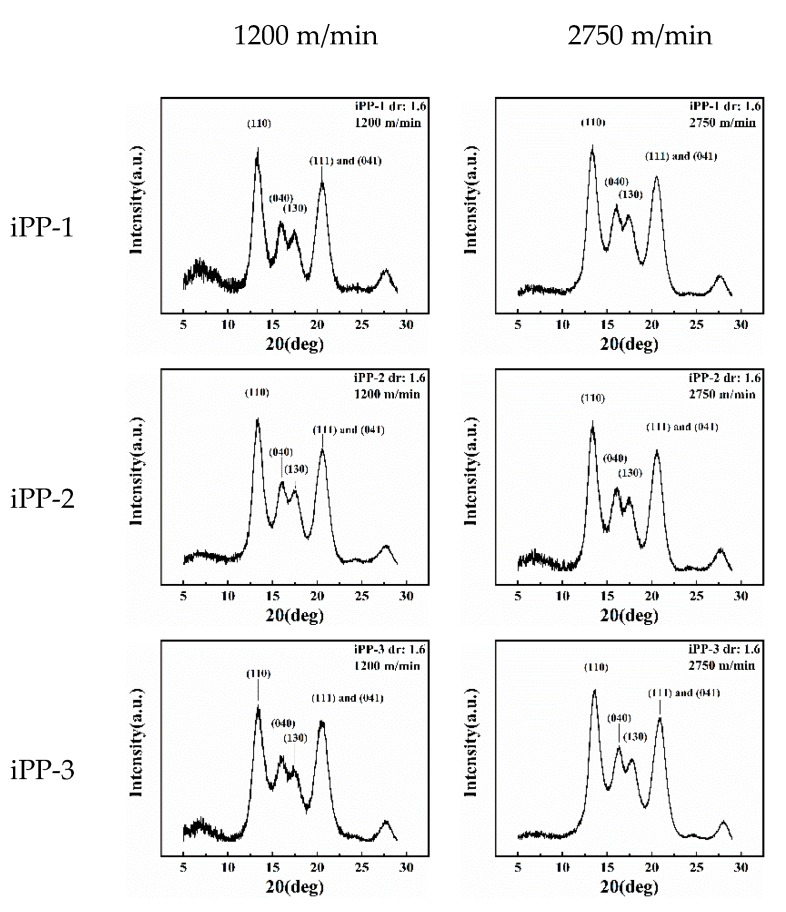
1D-WAXD curves of drawn iPP fibers (draw ratio: 1.6).

**Table 1 polymers-11-00729-t001:** ^13^C NMR analysis of the iPP samples.

Sample	*mmmm*%	*mmmr*%	*rmmr*%	*mmrr*%	*mmrm* + *rmrr*%	*rmrm*%	*rrrr*%
iPP-1	98.98%	0	0.28%	0.74%	0	0	0
iPP-2	97.10%	1.87%	0	0.75%	0.28%	0	0
iPP-3	93.64%	2.85%	0.81%	2.1%	0	0.4%	0.2%

**Table 2 polymers-11-00729-t002:** Melting temperatures (*T*_m_), melting enthalpies (Δ*H*_m_), and crystallinity indices (*χ*_c_) of the as-prepared powders of the iPP samples.

Sample	iPP-1	iPP-2	iPP-3
*T*_m_ (°C)	152.0	163.0	162.7
Δ*H*_m_ (J/g)	74.27	73.14	71.21
*χ*_c_ (%)	51.2	50.4	49.1

**Table 3 polymers-11-00729-t003:** Data of TREF test.

Sample	iPP-1	iPP-2	iPP-3
Soluble Fractions (%)	0.3	0.9	1.3

**Table 4 polymers-11-00729-t004:** Mechanical properties of the iPP samples.

Sample	Tensile Strength (MPa)	Flexural Modulus (MPa)	Simply Supported Beam Impact Strength (kJ/cm^2^)	Elongation at Break (%)
iPP-1	34.7	1654	2.15	16.21
iPP-2	35.3	1668	1.91	15.87
iPP-3	35.3	1496	1.89	13.48

**Table 5 polymers-11-00729-t005:** Processing properties of the iPP samples.

Sample	iPP-1	iPP-2	iPP-3
MFR (g 10/min)	34	36	35
Ash content (ppm)	180	237	220

**Table 6 polymers-11-00729-t006:** Tensile properties of as-spun iPP fibers.

Sample	Take-up Speed (m/min)	Line Density (dtex)	Tenacity (cN/dtex)	Modulus (cN/dtex)	Elongation at Break (%)
iPP-1	1200	97.5	1.69	12.0	242.5
	2100	91.8	1.37	16.4	157.1
	2750	88.9	1.22	15.6	120.0
iPP-2	1200	84.0	1.77	12.8	460.8
	2100	91.8	1.60	16.8	207.3
	2750	86.8	1.67	17.1	165.7
iPP-3	1200	98.8	2.06	16.0	165.5
	2100	92.3	1.37	15.8	106.4
	2750	77.9	2.36	25.0	69.6

**Table 7 polymers-11-00729-t007:** The tensile properties of drawn iPP fibers (draw ratio: 1.6).

Sample	Take-up Speed (m/min)	Line Density (dtex)	Tenacity (cN/dtex)	Modulus (cN/dtex)	Elongation at Break (%)
iPP-1	1200	52.6	2.98	21.1	28.0
	2100	56.2	2.63	18.0	30.8
	2750	53.6	2.57	19.9	19.8
iPP-2	1200	54.0	2.47	15.7	91.1
	2100	56.5	2.47	13.8	78.0
	2750	51.9	3.45	22.4	34.1
iPP-3	1200	59.6	3.77	28.9	23.9
	2100	55.2	3.02	23.4	22.5
	2750	71.2	2.84	22.5	25.5

**Table 8 polymers-11-00729-t008:** Sub-peaks derived from DSC curves for as-spun iPP fibers.

Sample	Take-up Speed (m/min)	*T*_m,1_ (°C)	*T*_m,2_ (°C)	*T*_m,3_ (°C)	*f* (R^2^)
iPP-1	1200	160.3	151.3	148.7	99.82%
	2100	159.8	152.4	148.5	99.88%
	2750	160.6	153.4	151.4	99.94%
iPP-2	1200	173.4	165.8	n/a	99.56%
	2100	171.4	163.9	n/a	99.02%
	2750	171.8	163.1	n/a	99.40%
iPP-3	1200	171.1	163.8	n/a	99.66%
	2100	172.7	165.2	n/a	99.87%
	2750	171.0	162.4	n/a	99.84%

**Table 9 polymers-11-00729-t009:** Sub-peaks derived from DSC curves for drawn iPP fibers.

Sample	Take-up Speed (m/min)	*T*_m,1_ (°C)	*T*_m,2_ (°C)	*T*_m,3_ (°C)	*f* (R^2^)
iPP-1	1200	162.4	153.4	149.1	99.90%
	2100	161.7	156.8	151.5	99.47%
	2750	161.2	153.1	143.8	99.95%
iPP-2	1200	173.4	165.8	n/a	99.56%
	2100	171.4	163.9	n/a	99.02%
	2750	171.8	163.1	n/a	99.40%
iPP-3	1200	174.2	166.4	n/a	99.85%
	2100	174.9	165.4	n/a	99.85%
	2750	171.3	162.5	n/a	99.49%

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
