# Peer review of "Structure and Properties of a Metallocene Polypropylene Resin with Low Melting Temperature for Melt Spinning Fiber Application"

_polymers, 2019, doi:10.3390/polym11040729_

Round 1

Reviewer 1 Report

While dealing with an interesting subject, the paper appears to reproduce the work of Bond and Spruiell from 2001 largely (see J. Appl. Polym. Sci. 82, 2001, 3223–3236 & 3237–3247). As a minimum requirement, these two papers also comparing smelt spinning behaviour and fibre properties of ZN-PP and MC-PP grades (actually even with a wider variation of materials and processing conditions) need to be referenced in the introduction, and the similarities and/or differences in findings should be discussed. The number of figures is definitely too high – nobody really needs that many WAXD curves and patterns (in that respect it should also be mentioned that the references on polymorphism are both thin and somewhat outdated). Finally, the language needs improvement, as already the first sentence of the abstract contains several mistakes (it should read “A newly developed metallocene-based isotactic polypropylene (iPP-1) resin with low melting temperature (Tm) is investigated for melt-spun fiber applications.”

Author Response

1) Comments: While dealing with an interesting subject, the paper appears to reproduce the work of Bond and Spruiell from 2001 largely (see J. Appl. Polym. Sci. 82, 2001, 3223–3236 & 3237–3247). As a minimum requirement, these two papers also comparing smelt spinning behaviour and fibre properties of ZN-PP and MC-PP grades (actually even with a wider variation of materials and processing conditions) need to be referenced in the introduction, and the similarities and/or differences in findings should be discussed.

Response:

The two suggested papers have been referenced in the introduction. But the two suggested papers emphasis points were fiber formation and the influencing factors of fiber properties, which belong to theoretical research. However, this manuscript is focused on a newly metallocene polypropylene for melt spinning fiber application, and compared to commercial iPP was to illustrate the application potential of the newly developed polypropylene.

2) Comments: The number of figures is definitely too high – nobody really needs that many WAXD curves and patterns (in that respect it should also be mentioned that the references on polymorphism are both thin and somewhat outdated).

Response:

    Thanks for the comments. We have put some of WAXD figures to Supporting Information. The outdate references have been replaced as well.

3) Comments: Finally, the language needs improvement, as already the first sentence of the abstract contains several mistakes (it should read “A newly developed metallocene-based isotactic polypropylene (iPP-1) resin with low melting temperature (Tm) is investigated for melt-spun fiber applications.”

Response:

      We have checked the manuscript carefully, and some mistakes have been revised.

Reviewer 2 Report

This manuscript by Wang presents a careful and thorough analysis of polypropylene resin using DSC, POM, TREF, and 2D XAXD. However, there is no information about metallocene catalysts for the production of isotactic polymers (iPP-1, iPP-2, and iPP3). The authors should be described which catalysts was used to synthesis the polymers. Additionally, the authors should be explained a possibility of enantiomorphic site control in the synthesis of iPP-1. If iPP-1 has stereo- and/or regiodefects in the polymer chain, the melting point (Tm) could be lower temperature compared with common isotactic polypropylenes. Therefore, I strongly recommend the discussion for these defects in polymer samples. After above aspects are fully addressed, I think this manuscript can be considered for the publication on Polymers.

Author Response

1) Comments: This manuscript by Wang presents a careful and thorough analysis of polypropylene resin using DSC, POM, TREF, and 2D XAXD. However, there is no information about metallocene catalysts for the production of isotactic polymers (iPP-1, iPP-2, and iPP-3). The authors should be described which catalysts was used to synthesis the polymers.

Response:

The metallocene isotactic polypropylene (iPP-1) was prepared using C2-symmetric metallocene catalyst. The metallocene catalyst for polypropylene was rac-Me2Si(2-Me-4-Ph-Ind)2ZrCl2, which was synthesized according to reference (Organometallics, 1994, 13, 954-963). iPP-2 and iPP-3 were commercial isotactic polypropylene.

2) Comments: Additionally, the authors should be explained a possibility of enantiomorphic site control in the synthesis of iPP-1. If iPP-1 has stereo- and/or region-defects in the polymer chain, the melting point (Tm) could be lower temperature compared with common isotactic polypropylenes. Therefore, I strongly recommend the discussion for these defects in polymer samples.

Response:

        The metallocene catalyzed i-PP, may possess different types of microstructural defects, such as stereo-defects and region-defects in the polymer chain. The stereo-defects are mainly rr isolated triads, while the region-defects are mainly 2,1 and 3,1 insertions. The insertion of regio-irregular units in the main chain lead to random distribution of microstructural defects. These defects will lower the thickness of lamellar crystal of polymer chain, which result into a lower melting temperature. The discussion has been added in the manuscript. 

Reviewer 3 Report

1)     It’s generally assumed that the melting point of the isotactic polypropylene increases by increasing the stereoregularity. In this work is reported the odd occurrence of a decrease of the melting temperature by increasing the regularity of the polymer chain. Can the authors justify it?

2)     The sample of i-PP from Exxon is called “degradable grade” what means such a definition? Is this polypropylene chemically different from usual?

3)     “The quick formation and growing of the crystal domains owing to the addition of calcium stearate as nucleating agent result into the distorted spherulite domains.” This statement in lines 200-201 possibly could help to answer to the point 1 I raised above, but actually the addition of calcium stearate concerns the only sample iPP-1?

4)     In my opinion no discussion can be made about the comparison of samples that the readers (and the reviewers) cannot test in their laboratory due to their substantial non-availability, being unknow the way they were produced.

Beside the other point I consider point 4 as a relevant obstacle to giving my agreement to publish this paper.

Author Response

1) Comments: It’s generally assumed that the melting point of the isotactic polypropylene increases by increasing the stereoregularity. In this work is reported the odd occurrence of a decrease of the melting temperature by increasing the regularity of the polymer chain. Can the authors justify it?

Response:

The metallocene made i-PP, characterized by chains including different types of microstructural defects, such as stereo-defects and region-defects. The stereo-defects are mainly rr isolated triads, while the region-defects are mainly 2,1 and 3,1 insertions. The insertion of regio-irregular units in the main chain lead to random distribution of microstructural defects. These defects will lower the thickness of lamellar crystal of polymer chain, which result into a lower melting temperature. The discussion has been added in the manuscript. Apparently, avoiding the region-defects may effectively improve the melting point of i-PP.  

2) Comments: The sample of i-PP from Exxon is called “degradable grade” what means such a definition? Is this polypropylene chemically different from usual?

Response:

Degraded iPP is based on high molecular weight conventional iPP, which is obtained by adding a peroxide degrading agent to reduce the molecular weight and molecular weight distribution. “Degradable” is misleading here, and it is removed from the manuscript.

3) Comments: “The quick formation and growing of the crystal domains owing to the addition of calcium stearate as nucleating agent result into the distorted spherulite domains.” This statement in lines 200-201 possibly could help to answer to the point 1 I raised above, but actually the addition of calcium stearate concerns the only sample iPP-1?

Response:

Calcium stearate is usually added as an acid scavenger to remove the acid from the catalyst and prevent corrosion of the processing equipment. In present case, calcium stearate is added in i-PP1. For the other two commercial products, we could not know the exact processing agents added for the business secret.  

4) Comments: In my opinion no discussion can be made about the comparison of samples that the readers (and the reviewers) cannot test in their laboratory due to their substantial non-availability, being unknow the way they were produced.

Response:

        iPP-1 was prepared using C2-symmetric metallocene catalyst(rac-Me2Si(2-Me-4-Ph-Ind)2ZrCl2. iPP-2 and iPP-3 were commercial isotactic polypropylene. Among them, iPP-2 was commercial metallocene iPP, and iPP-3 was commercial Ziegler-Natta iPP.

Round 2

Reviewer 3 Report

Dear authors, I am sorry but the reasons of my negative opinion remain because actually too  uncertain there is about the nature of the samples of polymer you have studied: from the catalyst used for the synthesis to the presence of calcium stearate or other scavanger in the commercial polypropylene iPP-2 and i-PP3. You just cited the calcium stearate to justify the different behavior of the samples. The guidelines of the researchers should take into account the reproducibility of the experiments they report by defining the source of the samples they are comparing 

Author Response

Comments: Dear authors, I am sorry but the reasons of my negative opinion remain because actually too uncertain there is about the nature of the samples of polymer you have studied: from the catalyst used for the synthesis to the presence of calcium stearate or other scavanger in the commercial polypropylene iPP-2 and i-PP3. You just cited the calcium stearate to justify the different behavior of the samples. The guidelines of the researchers should take into account the reproducibility of the experiments they report by defining the source of the samples they are comparing.

Response:

   Isotactic polypropylene sample 1 (iPP-1) is synthesized from C2-symmetric metallocene catalyst (rac-Me2Si(2-Me-4-Ph-Ind)2ZrCl2) with a Mw of 147,000 and PDI of 2.5, which is a pilot product synthesized by China Petroleum Lanzhou Chemical Research Center. 500 ppm calcium stearate (Guangdong Wei Lin Na New Material Technology Co. Ltd., ≥98%), 750 ppm primary antioxidant (Irganox 1010) and 1500 ppm secondary antioxidant (Irgafos 168) (Basf Performance Chemicals (Shanghai) Co. Ltd.) were added into the iPP powder to prepare pellets by extruding. The other two commercial iPP samples are commercial products and their composition could not be obtained for the business concerns.